# Human Metapneumovirus Escapes NK Cell Recognition through the Downregulation of Stress-Induced Ligands for NKG2D

**DOI:** 10.3390/v12070781

**Published:** 2020-07-20

**Authors:** Mohammad Diab, Dominik Schmiedel, Einat Seidel, Eran Bacharach, Ofer Mandelboim

**Affiliations:** 1The Concern Foundation Laboratories at the Lautenberg Center for Immunology and Cancer Research, Institute for Medical Research Israel-Canada (IMRIC), Faculty of Medicine, Hebrew University Hadassah Medical School, Jerusalem 91120, Israel; mohammad.diab@mail.huji.ac.il (M.D.); dominik.schmiedel@weizmann.ac.il (D.S.); einat.seidel@mail.huji.ac.il (E.S.); 2Department of Cell Research and Immunology, Faculty of Life Sciences, Tel Aviv University, Tel Aviv 69978, Israel; EranBa@tauex.tau.ac.il

**Keywords:** Pneumoviridae family, human metapneumovirus (HMPV), natural killer (NK), natural killer group 2D (NKG2D), stress-induced ligands

## Abstract

The Pneumoviridae family includes human metapneumovirus (HMPV) and human orthopneumovirus, which is also known as a respiratory syncytial virus (HRSV). These are large enveloped, negative single-strand RNA viruses. HMPV and HRSV are the human members, which commonly infect children. HMPV, which was discovered in 2001, infects most children until the age of five, which causes an influenza-like illness. The interaction of this virus with immune cells is poorly understood. In this study, we show that HMPV evades natural killer (NK) cell attack by downregulating stress-induced ligands for the activating receptor NKG2D including: Major histocompatibility complex (MHC) class I polypeptide-related sequences A and B (MICA, MICB), UL16 binding proteins ULBP2, and ULBP3, but not ULBP1. Mechanistically, we show that the viral protein G is involved in the downregulation of ULBP2 and that the viral protein M2.2 is required for MICA and MICB downregulation. These findings emphasize the importance of NK cells, in general, and NKG2D, in particular, in controlling HMPV infection, which opens new avenues for treating HMPV.

## 1. Introduction

Natural killer (NK) cells that belong to the innate immune system play an important role in the immune surveillance and elimination of transformed cells, virus-infected cells, bacteria, and fungi [1,2,3,4,5,6,7,8]. The killing activity of NK cells is regulated by several activating and inhibitory receptors [6,9]. The activating receptor family include, among others, the Natural Cytotoxicity Receptors (NCRs): NKp30, NKp44, and NKp46 (NCR1 in mice) and natural killer group 2D NKG2D. The NCR family of receptors recognize several ligands including pathogen-derived ones [6]. NKG2D recognizes ligands in which expression is induced following stresses such as heat shock, deoxyribonucleic acid (DNA) damage, transformation, and virus infection. Hence, the NKG2D ligands are known as stress induced-ligands [10,11,12]. In humans, eight different stress-induced ligands were identified, which include the major histocompatibility complex (MHC) class I polypeptide-related sequences A and B (MICA, MICB) and a family of UL16 binding proteins 1–6 (ULBP 1–6). The stress-induced ligands are differentially expressed on the cell surface under various conditions. However, all of them are recognized by one receptor NKG2D [13]. Therefore, it is not surprising to see that pathogens and tumors have developed several sophisticated strategies to escape NK cell recognition and NKG2D [14,15,16,17,18,19,20,21,22].

Human metapneumovirus (HMPV) is an emerging pneumovirus. It represents the first human member of the Metapneumovirus genus. HMPV is a ubiquitous respiratory pathogen, which is known to have circulated in the human populations for decades [23]. Serological evidence indicates that almost all the children under the age of five years are infected with HMPV [23]. It is a leading cause of acute upper and lower respiratory tract infections in children, infants, adults, and immune-compromised patients [24,25]. Epidemiological studies show that HMPV induces clinical symptoms ranging from mild disease to severe bronchiolitis and pneumonia [26,27,28,29], and, in some cases, could lead to death [30,31].

Only limited studies were performed to understand the interactions of HMPV with the immune system [32,33,34,35,36,37,38,39,40,41,42]. We have recently shown that HMPV infection induced expression of an unknown ligand, which was recognized by the activating human receptor NKp46 and its orthologue NCR1 in mice [32]. In addition, we showed that the infected cells sense HMPV infection via retinoic acid-inducible gene I(RIG-I) and upregulate CEACAM1 expression to inhibit viral growth [43]. In this case, we examined whether NKG2D ligands are affected following HMPV infection.

## 2. Materials and Methods

### 2.1. Virus Growth and Cell Lines

Vero, kidney epithelial cells derived from an African green monkey and A549, which are human alveolar type II-like epithelial cells, were maintained in Dulbecco’s modified Eagle medium (DMEM, Sigma-Aldrich, Saint Louis, MO, USA) with standard supplements and 10% fetal calf serum (FCS, Biological industries, Kibbutz Beit Haemek, Israel). To propagate HMPV, around 70% of confluent Vero cell monolayers were infected with recombinant (HMPV-GFP) strains, as previously described [43,44], using MOI 0.1 in medium containing 0.25 mg/mL trypsin for one hour. DMEM medium with standard supplements containing 3% FCS was then added. Five days after infection, cells were scraped from the plates, collected, freeze-thawed (−80 °C/37 °C) three times, and crude viruses were stored at –80 °C. Viral loads were determined by using quantitative RT-PCR (qRT-PCR, Life Technologies, Carlsbad, CA, USA) and/or by using fluorescence microscopy. For A549 infection experiments, around 70% of confluent cell monolayers were infected with MOI 3 of HMPV in serum-free media with 0.25 mg/mL trypsin for 1 h incubation at 37 °C. Then it was maintained in DMEM medium with standard supplements containing 2% FCS. HMPV/WT and HMPV/ΔG viruses efficiently infect A549 and Vero cells and infection rates were around 100%. In each experiment, virus titers were calibrated to ensure the same number of infected cells including both HMPV/WT and HMPV/ΔG.

### 2.2. Flow Cytometry and Antibodies 

Cells in the flow cytometry experiment were incubated on ice for 1 h with 0.3 µg of Ab per 100,000 cells. Detection was done with a secondary antibody staining for 30 min on ice. The analysis was performed using the FACS-Calibur flow cytometer (BD Biosciences, San Jose, CA, USA) and CellQuest software v4.1.

The following primary antibodies were used for flow cytometry: anti-MICA (clone 159227, R&D Systems, Minneapolis, MN, USA), anti-MICB (clone 236511, R&D Systems), anti-ULBP1 (clone 170818, R&D Systems), anti-ULBP2/5/6 (clone 165903, R&D Systems), anti-ULBP3 (clone 166514, R&D Systems), anti-HA tag (12CA5). In addition, the goat anti-mouse coupled to AlexaFluor 647 from (Jackson ImmunoResearch, Philadelphia, PA, USA) was used as the secondary antibody. 

The following antibodies were used for blocking assays: anti-NKG2D (clone 149810, R&D Systems) and anti-CD99 (12E7), which was used as an isotype control. Anti-CD107a (LAMP-1) was purchased from (BioLegend, Way San Diego, CA, USA: catalog number 328620). Anti-CD56 (Becton Dickinson, Franklin Lakes, NJ, USA) and anti-CD3 (BioLegend) antibodies were used to determine NK cell purity.

The following primary antibodies were used for Western blotting: anti-HA tag (clone 3F10, Roche, Basel, Switzerland), anti-ULBP2/RAET1H (catalog number LS-C409800-100, LifeSpan BioSciences, Seattle, WA, USA), and anti-vinculin (clone EPR8185, Abcam). The following secondary antibodies were used for Western blotting: anti-mouse-HRP, anti-rat-HRP, and anti-rabbit-HRP, which were all purchased from Jackson laboratories.

### 2.3. Primers and Lentiviral Constructs Used for Cloning the HMPV Proteins

The HMPV proteins: fusion protein (F), phosphoprotein (P), nucleoprotein (N), attachment protein (G), matrix proteins (M), (M2.1), and (M2.2) were amplified from cDNA derived from Vero cells infected with HMPV. This cDNA was cloned into the lentiviral vector pHAGE-DsRED(−)eGFP(+), which also contains green fluorescent protein GFP [32]. HA-tag was introduced into the C-terminus of the HMPV proteins. The transduction of each HMPV protein was around 100%. For the introduction of an HA tag into the HMPV genes, the lentiviral vectors described above were used as templates for further rounds of PCR and cloning, using the following primers: (All primers are listed 5′ to 3′):

For HMPV hemagglutinin (HA)-tagged G gene: G-Fw: 5’GCGCGGCCGCGCCGCCACCATGGAAGTAAGAGTGGAGAAC ‘3 

G-Rev-HA-tag 5’ GCCTCGAGctaAGCGTAATCTGGAACATCGTATGGGTATGCATGGTGTGGTGGGGAGCTT ‘3

For HMPV HA-tagged F gene: F-Fw: 5’ GCGCGGCCGCGCCGCCACCATGTCTTGGAAAGTGGTGATC ‘3

F-Rev-HA-tag 5’ CCGGATCCCTAAGCGTAATCTGGAACATCGTATGGGTAACTGTGTGGTATGAAGCCATT ‘3

For HMPV HA-tagged N gene: N-Fw: 5’ GCGCGGCCGCGCCGCCACCATGTCTCTTCAAGGGA TTCAC ‘3

N-Rev-HA-tag 5’ CCGGATCCTTAAGCGTAATCTGGAACATCGTATGGGTACTCATAATCATTTTGACTGTCG ‘3

For HMPV HA-tagged P gene: P-Fw: 5’ GCGCGGCCGCGCCGCCACCATGTCATTCCCTGAAGGAAAAGATAT ‘3

P-Rev-HA-tag 5’ GCCTCGAGCTAAGCGTAATCTGGAACATCGTATGGGTACATAATTAACTGGTAAATGTC ‘3

For HMPV HA-tagged M gene: M-Fw: 5’ GCGCGGCCGCGCC GCCACCATGGAGTCCTATCTGGTAGAC ‘3

M-Rev-HA-tag 5’ CCGGATCCTTAAGCGTAATCTGGAACATCGTATGGGTATCTGGACTTCAGCACATAT ‘3

For HMPV HA-tagged M2.1 gene: M2.1-Fw: 5’ GCGCGGCCGCGCCGCCACCATGTCTCGCAAGGCTCCATGC ‘3

M2.1-Rev-HA-tag 5’ CCGGATCCTCAAGCGTAATCTGGAACATCGTATGGGTACTGCACTTGATTAATGCTTTCAC ‘3

For HMPV HA-tagged M2.2 gene: M2.2-Fw: 5’ GCGCGGCCGCGCCGCCACCATGACTCTTCATATGCCTTGC ‘3

M2.1-Rev-HA-tag 5’ CCCTCGAGCTAAGCGTAATCTGGAACATCGTATGGGTAACTTAAGTAAGTTTTAACATATA ‘3

### 2.4. CD107a Degranulation Assay 

A total of 50 × 103 IL-2-activated bulk primary human NK cells were isolated using the EasySepTM human NK cell enrichment kit (StemCell Technologies, Vancouver, British Columbia, Canada). The NK cell purity was >90%. NK cells were incubated with anti-NKG2D mAb or with anti-CD99 (12E7) Ab (R&D Systems) for one hour on ice, which was followed by incubation with the A549 HMPV-infected cell line at an effector to target (E:T) of 1:1. Target cells were inoculated with the HMPV virus for 24 h before the assay. The allophycocyanin-conjugated anti-CD107a mAb and the PE conjugated anti-CD56 mAb (BioLegend) were added before adding the targets. All the mixtures were incubated for 2 h at 37 °C. 

The analysis of CD107a surface expression was performed on CD56+ cells, as described [45], and determined by FACS-Calibur flow cytometer. 

### 2.5. RNA Extraction and cDNA Preparation

Total RNA was isolated by using the Quick RNA Miniprep kit (Zymo Research, Irvine, CA, USA), according to the manufacturer’s instructions. RNA was reverse transcribed with Moloney murine leukemia virus (M-MLV) reverse transcriptase (Invitrogen, Carlsbad, CA, USA) and with polyT primers (Thermo Scientific/Fermentas, Waltham, MA, USA). Quantitative amplification was conducted on a PCR performed on a QuantStudio12K Flex Real Time PCR System (Life Technologies, Carlsbad, CA, USA) with gene-specific primers and Platinum SYBR Green qPCR SuperMix-UDG with ROX (Invitrogen). A newly synthesized transcript analysis was performed as previously described [20]. RNA was reverse transcribed and analyzed by real-time PCR.

### 2.6. Quantitative Real-Time PCR

Quantitative PCR was used to measure mRNA expression as follows. cDNA was mixed with 150 µM of both the forward and reverse primers in a final volume of 5 µL and mixed with 5 µL of SYBR Green qPCR SuperMix-UDG with ROX (Invitrogen). Glyceraldehyde-3-phosphate dehydrogenase (GAPDH) and hypoxanthine phosphoribosyltransferase (HPRT) were used as endogenous reference genes for PCR quantification. PCR was performed on QuantStudio12K Flex Real Time PCR System (Life Technologies). Newly synthesized transcript analysis was performed as previously described [20].

The following primers were used (listed from 5′ to 3′): 

GAPDH forward, GAGTCAACGGATTTGGTCGT, GAPDH reverse, GATCTCGCTCCTGGAAGATG, HPRT forward, TGACACTGGCAAAACAATGCA, HPRT reverse, GGTCCTTTTCACCAGCAAGCT, MICA forward, ATCTTCCCTTTTGCACCTCC, MICA reverse, AACCCTGACTGCACAGATCC, MICB forward, CTGCTGTTTCTGGCCGTC, MICB reverse, ACAGATCCATCCTGGGACAG, ULBP1 forward, GCGTTCCTTCTGTGCCTC, ULBP1 reverse, GGCCTTGAACTTCACACCAC, ULBP2 forward, CCCTGGGGAAGAAACTAAATGTC, ULBP2 reverse, ACTGAACTGCCAAGATCCACTGC, ULBP3 forward, AGATGCCTGGGGAAAACAACTG, ULBP3 reverse, GTATCCATCGGCTTCACACTCAC

### 2.7. Western Blot Analysis

Cells were counted and lysed in buffer containing 0.6% sodium dodecyl sulfate (SDS)and ten mM Tris (pH 7.4). The proteins’ concentration in each sample was determined by the Bradford method. Equal volumes of samples containing equal protein concentrations were then run on 12.5% Tris-HCl gels, then transferred onto nitrocellulose membrane from (Bio-rad, Hercules, CA, USA), and incubated for 1 h with 5% milk. Membranes were incubated overnight with primary Abs. Abs were diluted in 5% bovine serum albumin (BSA) in phosphate-buffered saline (PBS). Chemiluminescence by Ab-linked horseradish peroxidase (HRP) (Jackson ImmunoResearch, Philadelphia, PA, USA) was detected using an enhanced chemiluminescence (EZ-ECL) detection kit (Biological Industries, Kibbutz Beit Haemek, Israel). The Image Lab software v6.0.1 was used for quantification.

### 2.8. Proteasomal and Lysosomal Inhibitors

For proteasome and lysosome inhibition, cells were mock-infected or infected with the HMPV variants HMPV/WT and HMPV/ΔG for 18 h and then incubated with the proteasomal inhibitor epoxomicin 50 ng/mL, (Merck Millipore, Darmstadt, Germany) or with the lysosomal inhibitor leupeptin 10 µg/mL (Merck Millipore) for an additional 6 h. As controls, cells were left untreated or treated with equivalent concentrations of double-distilled water (DDW) or dimethyl sulfoxide (DMSO, epoxomicin solution).

### 2.9. Statistical Analysis

The *n* value within the figures refers to the biological replicates number and is indicated in the respective figure legends. Figure 1 and Figure 3 were analyzed using one-way ANOVA for each ligand expression, which was followed by the post-hoc test to identify significant differences in NKG2D ligands’ expression between multiple groups means of mock-infected, HMPV/WT, and HMPV/ΔG-infected cells groups. A corrected *p*-value < 0.0167 (0.05/3 [number or groups]) considered statistically significant *p* values were estimated and indicated in the respective figure legends. Figure 2 and Figure 4 were analyzed using two-way ANOVA, which was followed by the post-hoc test. A Bonferroni adjustment was performed for multiple comparisons. A corrected *p*-value < 0.0167 was considered statistically significant. *p* values were estimated and indicated in the respective figure legends.

## 3. Results

### 3.1. Ligands of NKG2D Receptor are Downregulated Following HMPV Infection Influencing NKG2D-Mediated Killing

We have previously shown that HMPV infection affects the expression of an unknown NKp46 ligand [32]. To investigate if NKG2D ligands are affected by HMPV, we infected A549 cells (human cell line that constitutively expresses NKG2D ligands and can be efficiently infected with this virus) with recombinant HMPV expressing green fluorescent protein GFP (HMPV/WT) at MOI 3 [32,43,46] (Figure 1). The infected cells were identified as GFP-positive cells, and the infection rates were around 100%. Twenty-four hours following infection, we stained the mock-infected and the infected cells for the expression of NKG2D ligands: MICA, MICB, ULBP1, ULBP2, ULBP3, and ULBP4. We observed a significant reduction of MICA, MICB, ULBP2, and ULBP3, but not ULBP1 (Figure 1A, quantified in Figure 1C). ULBP4 is not expressed on A549 cells. We also investigated NKG2D ligands during the infection with HPMV, which lacked the G protein (HMPV/ΔG) since this recombinant virus has been shown to upregulate the expression of an unknown NKp46 ligand [32]. For this purpose, we infect the same cells with HMPV/ΔG at MOI 3 (infection rates around 100%). MICA, MICB, and ULBP3 were still downregulated (Figure 1B, quantified in Figure 1C). However, ULBP2 was not (Figure 1b, quantified in Figure 1C). These findings indicate that HMPV targeted specific ligands of the activating NKG2D and that the G protein of HMPV is involved in the downregulation of ULBP2 only.

The CD107a glycoprotein is located on the outer membrane of lytic granules, which are found inside the cell and is not detectable (or have a low level of expression) on the cell surface of non-activated NK cells. Upon activation, the lytic granules move to the cell surface. During this process, the lytic content of granules, which include granzymes and perforin, is exocytosed and the CD107 (LAMP-1) molecules temporarily appear on the cell surface [47]. To test whether the NKG2D ligands’ downregulation will affect NK activity, we performed NK CD107a degranulation assays. A549 cells were infected with the HMPV/WT or HMPV/ΔG virus (the infection rates were around 100%). Twenty-four hours later, the infected cells were incubated with primary IL-2-activated bulk human NK cells in the presence or absence of the anti-NKG2D antibody, and the expression of CD107a on the surface of the NK cells was determined. A549 is a tumor cell line that constitutively expresses stress-induced ligands for the NKG2D receptor. Thus, as can be seen in Figure 2, upon incubation of mock-infected cells with NK cells, an increase of CD107a expression is observed. In the absence of the blocking antibody, significantly less CD107a was detected on the NK cells incubated with HMPV/WT-infected cells likely because the infected cells express lower expression of NKG2D ligands compared to mock-infected cells (Figure 1). When cells were infected with HMPV/ΔG, the CD107a expression was partially restored since the ULBP2 expression of the HMPV/ΔG-infected cells was not downregulated (Figure 1). These effects were due to NKG2D interactions with its ligands since blocking of NKG2D resulted in an equivalent significant decrease of CD107a expression in all cells (Figure 2).

### 3.2. Mechanism of ULBP2 Downregulation

To characterize the mechanisms responsible for NKG2D-ligands downregulation, we tested the mRNA levels, using qRT-PCR of each of the downregulated ligands, following HMPV/WT and HMPV/ΔG infection. As can be seen in Figure 3, at 24-h post-infection, the mRNA levels of all ligands whom expression was downregulated on the cell surface (Figure 1). MICA (Figure 3A), MICB (Figure 3B), ULBP2 (Figure 3D), and ULBP3 (Figure 3E) were upregulated following infection with both HMPV/WT and HMPV/ΔG. The UL16 binding protein-1 (ULBP1) mRNA levels remain unchanged (Figure 3C). 

Since, in the absence of G, no downregulation of ULBP2 was observed (Figure 1), and since an increase of ULBP2 mRNA expression was detected following HMPV infection (Figure 3D), we concentrated in studying the G-dependent mechanism of ULBP2 downregulation. We first determine the total level of the ULBP2 ligand in total cell lysates (Figure 4). In agreement with the FACS results (Figure 1), infection of A549 with HMPV/WT significantly decreases the ULBP2 protein level. Infection of the A549 cells with HMPV/ΔG increases the total protein level of ULBP2 (Figure 4A, quantified in Figure 4B). To analyze the mechanism by which HMPV affects ULBP2 expression, we infected the A549 cells with HMPV/WT or with HMPV/ΔG in the presence of either proteasomal (PI) or lysosomal inhibitors (LI). Treatment with both inhibitors increased the expression of ULBP2 in mock-infected cells and cells infected with HMPV/ΔG (Figure 4). Importantly, treatment with the PI epoxomicin did not affect the expression of ULBP2 following infection with HMPV/WT (Figure 4A, quantified in Figure 4B). However, elevated ULBP2 levels were observed when the LI leupeptin was used (Figure 4A, quantified in Figure 4B). These findings suggest that HMPV/WT-infected cells target the ULBP2 protein expression likely via its viral protein G to lysosomal degradation.

### 3.3. The HMPV Protein M2.2 Targets Major Histocompatibility Complex (MHC) Class I Polypeptide-Related Sequences A and B (MICA) and (MICB) 

The HMPV genome is comprised of a negative-sense single-stranded RNA, containing eight genes that encode for nine proteins. The order of the genes in the genome is 3’-N-P-M-F-M2-SH-G-L-5’. The proteins are: nucleoprotein (N), phosphoprotein (P), matrix protein (M), fusion glycoprotein (F), putative transcription factor (M2.1), RNA synthesis regulatory factor (M2.2), small hydrophobic glycoprotein (SH), attachment glycoprotein (G), and viral polymerase (L) [31]. To test if any of these proteins is responsible for the downregulation of the NKG2D ligands, we expressed all viral proteins labeled with an HA-tag in A549 cells. We succeeded in expressing all viral proteins except SH and L (see Western blot verifications in Figure 5A). The M2.2 protein is a small protein (8 kDa) that could not be detected using a Western blot and, therefore, the expression of this protein was verified by using intracellular Fluorescence-activated cell sorting (FACS) staining using a HA-specific antibody (Figure 5B). Staining of cells expressing the HMPV proteins or empty vector reveals almost no changes in the expression of MICA, MICB, ULBP1, ULBP2, and ULBP3, while expression of M2.2 was sufficient to reduce the expression of MICA and MICB. These results imply that the G protein is necessary for the downregulation of ULBP2 but not sufficient. Additionally, the downregulation of ULBP2 and ULBP3 occurred likely through a combination of more than one HMPV proteins.

## 4. Discussion

HMPV is one of the major pathogens that infect children. It causes acute respiratory tract infection, and, in severe cases, infection with this virus could be lethal [24,30,31,48,49]. To date, there is no effective vaccine or treatment against HMPV, and the contribution of the immune system against HMPV infection is poorly understood. In this case, we describe a novel immune evasion mechanism employed by HMPV. We show that HMPV tries to avoid NK cell killing by downregulating various stress-induced ligands of the NK-activating receptor NKG2D, specifically MICA, MICB, ULBP2, and ULBP3. These findings emphasize the important role played by NK cells in controlling HMPV infection.

We demonstrate that the infected cells respond to the infection and that, upon infection, the mRNA and the total protein amount of MICA, MICB, ULBP2, and ULBP3 are upregulated. This represents a rapid cellular response to the infection. The virus, however, developed a strategy to downregulate the surface expression of the ligands for NKG2D to avoid immune cell attack. Similar observations were noted in other viruses such as HCMV [50]. We show this downregulation affects the NK activity resulted in reduced NK cell recognition of the infected cells. NKG2D ligands, which are known to be activated on numerous stress pathways, their name’s stress-induced ligands are sensitive to heat shock, oxidative stress, genotoxic stress, and viral infection [51]. Moreover, proteasomal and lysosomal inhibitors were also shown to increase NKG2D ligands’ expression [52]. This study shows the same phenomena since UL16 binding protein-2 (ULBP2) expression is upregulated in treating proteasomal and lysosomal inhibitors. Mechanistically, our findings indicate that, in the infected cells, HMPV upregulates the transcript of ULBP2, while downregulating the surface expression of the ULBP2 protein. In addition, the observations that infection with the HMPV/ΔG virus increases the protein levels of ULBP2 and that ULBP2 is further upregulated in the presence of lysosome inhibitors, suggest that HMPV, via its viral protein G, contributes toward downregulating the surface expression of ULBP2, likely through the lysosomal pathway. However, the G protein by itself was not sufficient to affect ULBP2 surface expression, and, thus, we concluded that it acts together with additional protein(s) to downregulate ULPB2 expression.

It is still unknown how the infected cells sense the infection and react by upregulating the mRNAs of several NKG2D ligands and it is also unknown why the expression of ULBP1 is not changed. We previously showed that HMPV-infected cells sense the infection via the RIG-I-dependent pathway, which results in CEACAM1 upregulation [43]. Whether RIG-I was also involved in the upregulation of the stress-induced ligand will be investigated in the future.

We further demonstrated that, by overexpressing each of the HMPV viral proteins F, P, M2.1, M, N, G, and M2.2 individually, M2.2 was sufficient to downregulate MHC class I polypeptide-related sequences A and B (MICA) and (MICB) surface expression. The downregulation of ULBP2 and ULBP3 occurred likely through a combination of more than one HMPV protein.

In summary, we show in this study a new immune-evasion strategy developed by HMPV to escape NK cell recognition. We show that HMPV targets the surface expression of the NKG2D receptor ligands MICA, MICB, ULBP2, and ULBP3. The reduced surface expression of MICA and MICB was mediated by the HMPV viral protein M2.2, while the expression of ULBP2 was targeted to lysosomal degradation with the intervention of the HMPV viral protein G.

## Figures and Tables

**Figure 1 viruses-12-00781-f001:**
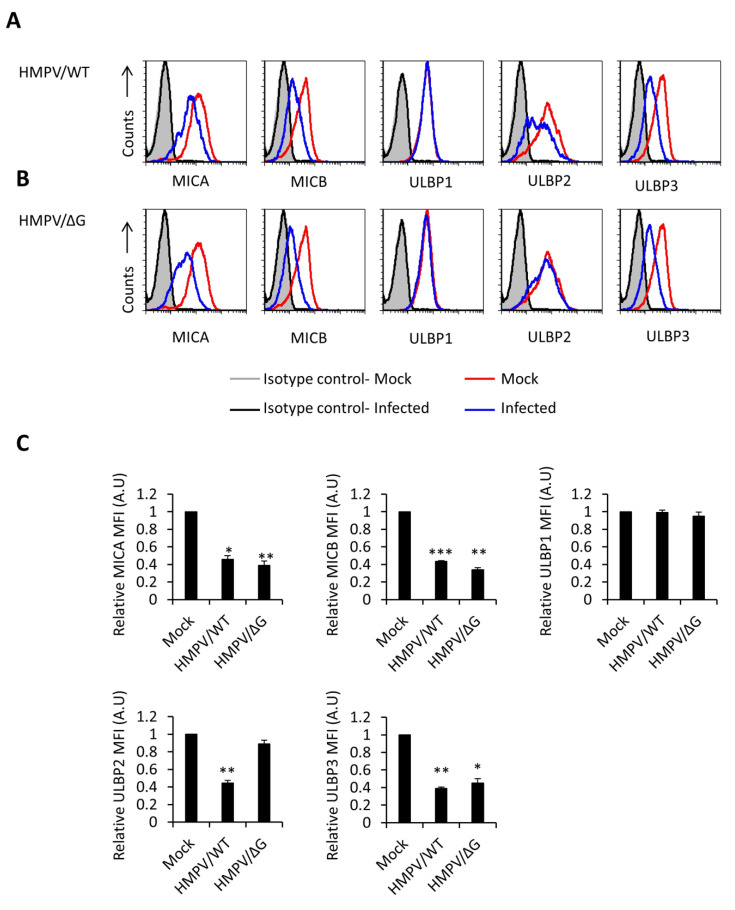
Infection of A549 cells with human metapneumovirus (HMPV) decreases the expression of NKG2D ligands. (**A** and **B**) Fluorescence-activated cell sorting (FACS) analysis of NKG2D ligands’ expression on the mock-infected A549 cells (empty red histogram) and on HMPV/Wilde Type (WT) (**A**) or HMPV/∆G (**B**)-infected A549 cells (empty blue histogram) at 24-h post-infection. The filled gray histogram and the empty black histogram represent the staining of the mock-infected and infected A549 cells with a control antibody, respectively. (**C**) Quantification of the expression of NKG2D ligands on mock-infected, HMPV/WT, and HMPV/∆G-infected cells. Shown is the mean fluorescence intensity (MFI) of stress-induced ligands on the infected cells relative to mock-infected cells (set as 1) from five independent experiments combined. Statistical analysis performed using one-way ANOVA (*n* = 5). *p* values were estimated using a post-hoc test. (*** *p* < 0.0001, ** *p* < 0.005, * *p* < 0.01).

**Figure 2 viruses-12-00781-f002:**
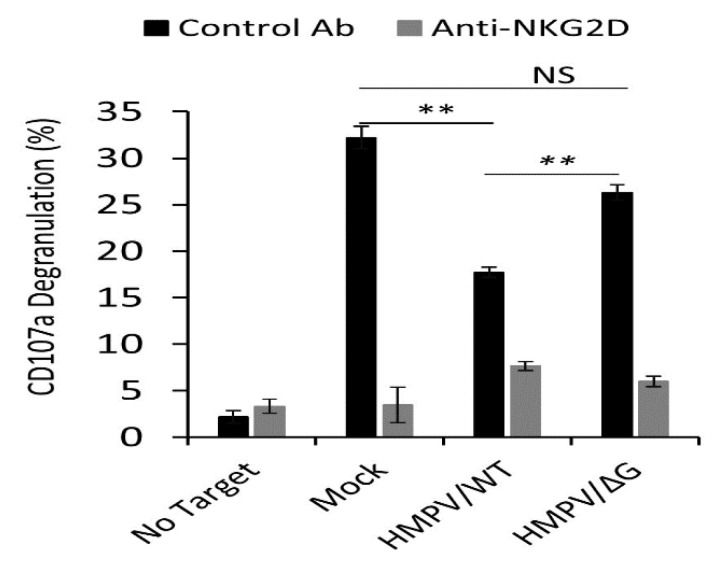
HMPV infection decreases natural killer (NK) cell activation. Primary IL-2-activated NK cells were incubated with the target cells, mock-infected A549 cells (Mock), HMPV/WT-infected A549 cells (HMPV/WT), and HMPV/∆G-infected A549 cells (HMPV/∆G) at a 1:1 ratio with or without blocking antibodies against the natural killer group 2D (NKG2D) receptor that were included during the infection period. CD107a expression was assessed. The experiment included two independent NK cell donors. The experiment without NKG2D blocking and with the blocking of anti-NKG2D were repeated three times. Statistical analysis was performed on all combined data using two-way ANOVA (*n* = 3). *p* values were estimated using a post-hoc test. ** *p* < 0.005. NS—Not significant.

**Figure 3 viruses-12-00781-f003:**
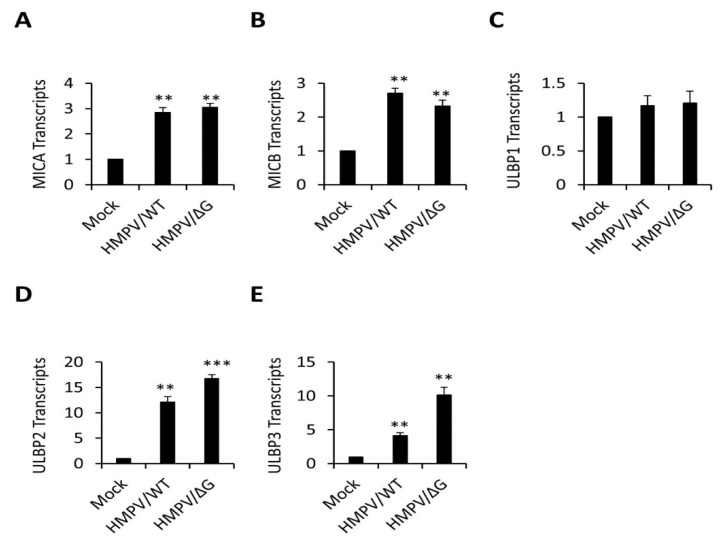
HMPV-infected A549 cells upregulate mRNA levels of natural killer group 2D (NKG2D) ligands. (**A**–**E**) Real Time PCR quantification of NKG2D ligands, major histocompatibility complex (MHC) class I polypeptide-related sequences A and B (MICA) (**A**), (MICB) (**B**), UL16 binding proteins (ULBP1) (**C**), (ULBP2) (**D**), and (ULBP3) (**E**) expression in mock-infected A549 cells (Mock), HMPV/WT-infected A549 cells (HMPV/WT), and HMPV/∆G-infected A549 cells (HMPV/∆G). The expression of NKG2D ligands in the infected cells was determined relative to mock-infected cells (Mock) in which the expression level was set up as 1. Data are representative of three independent experiments combined. Statistical analysis was performed using one-way ANOVA (*n* = 3). *p* values were estimated using a post-hoc test. *** *p* < 0.0001 and ** *p* < 0.005).

**Figure 4 viruses-12-00781-f004:**
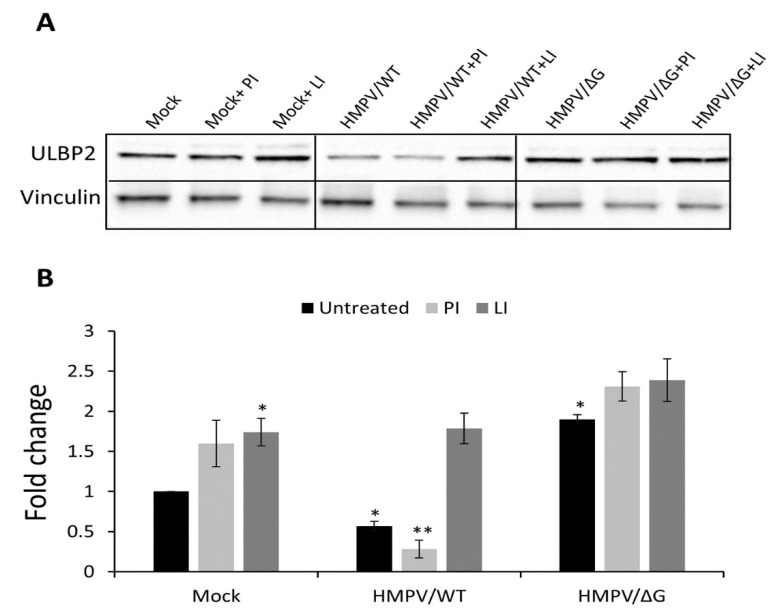
UL16 binding protein-2 (ULBP2) total expression following HMPV infection. (**A**) Western Blot analysis of ULPB2 (28 kDa) expression in mock-infected A549 cells (Mock), HMPV/WT-infected A549 cells (HMPV/WT), and HMPV/∆G-infected A549 cells (HMPV/∆G) with or without treatment with proteasome and lysosome inhibitors (PI, LI, respectively) (upper panel). Vinculin was used as a loading control (lower panel) (124 kDa). (**B**) Quantification of the relative ULBP2 protein levels shown in (**A**): protein quantity was normalized by vinculin expression, and then the ratio of ULBP2 levels for each treated cells was calculated as follows: (level in treated cells)/(level in mock-infected cells). The figure shows one representative experiment out of three independent experiments performed. Statistical analysis was calculated on the data from all experiments combined using two-way ANOVA (*n* = 3). *p* values were estimated using a post-hoc test. * *p* < 0.01 and ** *p* < 0.005.

**Figure 5 viruses-12-00781-f005:**
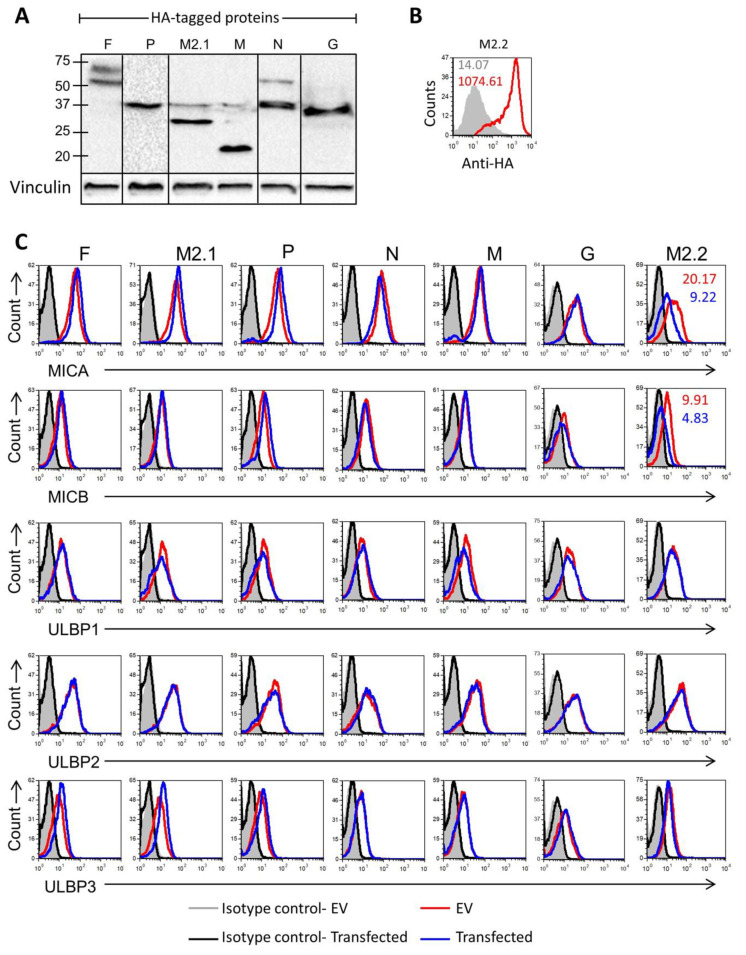
NKG2D ligands expression following expression of HMPV proteins. (**A**) Western blot analysis of the HMPV HA-tagged proteins F, P, M2.1, M, N, and G in transfected A549 cells (upper panels). Vinculin expression was used as a loading control (lower panels). Contrasts in the WB were altered for better clarity. (**B**) Fluorescence-activated cell sorting (FACS) analysis of intracellular staining with the anti-HA antibody of the HA-tagged protein M2.2 in transfected A549 cells. The red empty histogram represents HA staining, and a gray-filled histogram represents staining with a control antibody. Intracellular expression levels were also shown as MFI. The figure is representative of three independent experiments. (**C**) FACS analysis of the MHC) class I polypeptide-related sequences A and B (MICA), (MICB), UL16 binding proteins (ULBP1), (ULBP2), and (ULBP3) expression in A549 cells transfected with the indicated proteins. The blue empty histogram represents staining of the transfected cells. The empty red histogram represents staining of cells transfected with an empty vector. The empty black histogram and the gray-filled histogram are the staining of the HMPV protein-transfected cells and empty vector-transfected cells with a control antibody. Surface expression levels were also shown as mean fluorescence intensity (MFI) in the relative differences. The figure combines representative staining obtained in five independent experiments.

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
