# Peer review of "Human Metapneumovirus Escapes NK Cell Recognition through the Downregulation of Stress-Induced Ligands for NKG2D"

_viruses, 2020, doi:10.3390/v12070781_

Round 1

Reviewer 1 Report

Viruses-85815

This work explores an interesting aspect of the modulation of the immune response by HMPV. However, the manuscript presents conclusions that are not supported by the data. Statistical analysis needs to be appropriately applied. The quality of the manuscript would benefit from an additional round of editing. Specific comments are as follows:

  1. Figure 1- It was not clear the rationale for using the deltaG recombinant virus to study the expression of the NKG2D ligands. The authors did, though, a valid screening for the M2.2 protein (Figure 5C).
  1. Figure 1. The experiment is missing a positive control of the ligands’ expression.
  1. ANOVA statistical analysis should be applied since there are more than two conditions being compared. Post-test should be applied accordingly. The description of the statistical analysis should be included in methods.
  1. Figure 3. It needs to be stated the rationale for using both infections combined. In addition, controls of single infections WT and delta G must be included. Without those controls the data are difficult to be interpreted, or compared with data in figure 1C.
  1. Line 256 – “Treatment with both inhibitors increased the expression of ULBP2 
in MOCK uninfected cells …” . It should be mock-infected.
  1. Line 257 – “Treatment with both inhibitors increased the expression of ULBP2 
in MOCK uninfected cells and in cells infected with HMPV/ΔG…” This sentence implies that there was an increased expression with the ΔG +inhibitors, when this was NOT the case, unless that the statistical analysis/symbols are properly applied. This needs to be revised, otherwise is confusing.
  1. Line 260 – “While HMPV, via 
its viral protein G and probably additional proteins act to downregulate the surface expression of 
ULBP2 via the lysosomal pathway” This statement is not supported by the data. Based on the data in Fig. 4B, there is not experimental evidence that the G protein alters the expression of ULBP2 at the transcriptional level in the lysosome pathway. Additional experiments with the G protein expression would be needed to test that.
  1. Line 311. “We show that HMPV escapes NK cell killing…” This is an overstatement that is not supported by the data presented. HMPV-infected cells were degranulated, although some reduced efficiency was observed when compared with mock-infected cells.
  1. Line 322- “We further show that the G protein by itself was not sufficient to affect ULBP2 expression, and thus, we concluded that it acts together with additional protein(s) to downregulate 
ULPB2 expression”. It is not clear how this conclusion was reached. Could this effect be due to an alteration of a different pathway? or an inhibition of a different molecular strategy? In Figure 3, it is shown that the lack of G protein is sufficient to decrease the ULPB2 protein expression.
  1. Line 332 – “The downregulation of ULBP2 and ULBP3 occurred probably through a combination of more than 
one HMPV protein” How that conclusion was reached?
  1. Line 337 – “…the expression of ULBP2 targeted to lysosomal degradation with 
the intervention of the HMPV viral protein G” That statement is not supported by the data since it is not clear whether there was a significant change of ULBP2 expression between deltaG and deltaG+Li.
  1. A more complete discussion of the findings is needed.
  1. Additional round of editing will benefit the quality of the manuscript.

Author Response

REVIEWER#1

This work explores an interesting aspect of the modulation of the immune response by HMPV. However, the manuscript presents conclusions that are not supported by the data. Statistical analysis needs to be appropriately applied. The quality of the manuscript would benefit from an additional round of editing. Specific comments are as follows

  1. Figure 1- It was not clear the rationale for using the deltaG recombinant virus to study the expression of the NKG2D ligands. The authors did, though, a valid screening for the M2.2 protein (Figure 5C).

Response: We thank the reviewer for evaluating our manuscript. We have stated previously that HMPV/WT-infected cells and HMPV/ΔG-infected cells upregulate the ligand expression of the activating receptor NKp46 (ref 32 in the current study). Thus, it is important to study the relevance of other ligands in response to these recombinant viruses. Please see lines 181-182 and lines 189-192 in the manuscript.

  1. Figure 1. The experiment is missing a positive control of the ligands’ expression.

Response: We use the A549 cell line which is well established for expressing the NKG2D ligands. Please see Glasner et al., 2017. And we also emphasize that in the manuscript. Please see line 183.

  1. ANOVA statistical analysis should be applied since there are more than two conditions being compared. Post-test should be applied accordingly. The description of the statistical analysis should be included in methods.

Response: We agree with the reviewer, and we performed one-way and two-way ANOVA analyses. Followed by the post-hoc test. And the p values were estimated and indicated in each figure. Description of the statistical analysis included in the methods for all figures. See lines 179-188 in the revised manuscript.

  1. Figure 3. It needs to be stated the rationale for using both infections combined. In addition, controls of single infections WT and delta G must be included. Without those controls the data are difficult to be interpreted, or compared with data in figure 1C.

Response: I think there is a misunderstanding of the figure. The figure does not show both infections combined. We performed a single infection of WT virus and ΔG virus compared to mock-infected cells. Furthermore, this assay was performed at the same time with FACS staining of the surface expression of NKG2D ligands, indicating that the elevated mRNA levels occurred at the same time with the downregulation of the surface expression of NKG2D ligands.

  1. Line 256 – “Treatment with both inhibitors increased the expression of ULBP2 
in MOCK uninfected cells …” . It should be mock-infected.

Response: We agree with the reviewer and we corrected it to “mock-infected”. Please see line 258 in the revised manuscript.

  1. Line 257 – “Treatment with both inhibitors increased the expression of ULBP2 
in MOCK uninfected cells and in cells infected with HMPV/ΔG…” This sentence implies that there was an increased expression with the ΔG +inhibitors, when this was NOT the case, unless that the statistical analysis/symbols are properly applied. This needs to be revised, otherwise is confusing.

Response: Treatment with proteasome and lysosome inhibitors is a stressed condition to the cell. And one mechanism of the cellular response induced by this stress is to express the stress-induced ligands as ULBP2 as can be seen in mocked-infected cells (Mock+PI; Mock+LI) and  HMPV/ΔG-infected cells (HMPV/ΔG+PI; HMPV/ΔG+LI). However, due to the presence of the viral protein G in the HMPV/WT virus, the HMPV/WT-infected cells ca not induce the total levels of ULBP2 since the viral G protein maybe with other viral components send the surface expression of ULBP2 to lysosomal degradation. This description is proved when the lysosomal degradation is inhibited and the total levels recovered to the higher levels (of the stressed condition) similar to the ULBP2 expression in mocked-infected cells.

  1. Line 260 – “While HMPV, via 
its viral protein G and probably additional proteins act to downregulate the surface expression of 
ULBP2 via the lysosomal pathway” This statement is not supported by the data. Based on the data in Fig. 4B, there is not experimental evidence that the G protein alters the expression of ULBP2 at the transcriptional level in the lysosome pathway. Additional experiments with the G protein expression would be needed to test that.

Response: We agree with the reviewer. Thus, we carefully state that the effect of the viral protein G and probably with other viral proteins is to downregulate only the surface expression of the ULBP2 ligand via lysosomal degradation since the only difference between the HMPV/WT and HMPV/ΔG viruses is the presence of the viral  protein G as mentioned in the manuscript.

  1. Line 311. “We show that HMPV escapes NK cell killing…” This is an overstatement that is not supported by the data presented. HMPV-infected cells were degranulated, although some reduced efficiency was observed when compared with mock-infected cells.

Response: We agree with the reviewer. We corrected the statement to “We show that HMPV tries to avoid NK cell killing”. Please see line 314 in the revised manuscript.

  1. Line 322- “We further show that the G protein by itself was not sufficient to affect ULBP2 expression, and thus, we concluded that it acts together with additional protein(s) to downregulate 
ULPB2 expression”. It is not clear how this conclusion was reached. Could this effect be due to an alteration of a different pathway? or an inhibition of a different molecular strategy? In Figure 3, it is shown that the lack of G protein is sufficient to decrease the ULPB2 protein expression.

Response: Overexpression of the viral protein G does not affect the surface expression of the ULBP2 protein as can be seen in figure 5C. However, figure 1B, figure 2, and figure 4B support our conclusion that the viral protein G is essential for ULBP2 downregulation but its individual expression is not sufficient and probably needs other viral proteins.

  1. Line 332 – “The downregulation of ULBP2 and ULBP3 occurred probably through a combination of more than 
one HMPV protein” How that conclusion was reached?

Response: Overexpression of seven out of nine viral proteins does not reveal the relevant protein responsible for ULBP3 downregulation. It is still possible that the unexpressed SH or the viral polymerase L proteins are responsible for this downregulation. However, we think that a combination of the viral proteins is a possible idea.

  1. Line 337 – “…the expression of ULBP2 targeted to lysosomal degradation with 
the intervention of the HMPV viral protein G” That statement is not supported by the data since it is not clear whether there was a significant change of ULBP2 expression between deltaG and deltaG+Li.

Response: viral infection is thought to be another condition of stress to the cell. Thus, treatment with lysosome inhibitor or the viral infection seems to induce the expression of the stress-induced ligands as ULBP2 as you can see in Figur4A and 4B in (HMPV/ΔG and HMPV/ΔG+LI), and also by the transcriptional levels of ULBP2 (Figure 3). However, the involvement of the viral protein G was shown only in HMPV/WT infection, since inhibiting the lysosome prevents the ULBP2 degradation which activated by the G protein. Please see the new discussion in the revised manuscript.

  1. A more complete discussion of the findings is needed.

Response: We agree with the reviewer. Please see the new discussion in the revised manuscript.

  1. Additional round of editing will benefit the quality of the manuscript

Response: Response: We agree with the reviewer. Please see the revised manuscript.

Reviewer 2 Report

Dear editor:

I have carefully read this manuscript.

HMPV is a kind of virus which easily infect children during the age of 3 to 5. The mechanism of HMPV infection and pathogenesis was poorly understood. In this manuscript, Diab et al authors reported that HMPV could evade NK cell attack by downregulated the expression of the ligands of NKG2D on host cells, such as MICA, MICB, ULBP2, and ULBP3. This manuscript also provided some clues on the mechanism of HMPV evaded from NK cell, specifically, viral protein G was involved in the downregulation of ULBP2, and protein M2.2 was involved in the downregulation of MICA, MICB.

Overally, the work is well done. I suggest acceptance of this manuscript with some minor revisions.

Some questions:

Perhaps the most surprising finding is HMPV could evade NK cell attack by downregulated the expression of several ligand of NKG2D on host cells. But here the author should describe the “mock” clearly in almost all figures.

The HMPV/∆G was used to infect cells, it is not clear to this reviewer why the authors used this viral strain in the first place, why not other eight viral proteins delete strain or by a screen.

The authors described that protein G played an important role in regulating the expression of ULBP2. ULBP2 or other NKG2D ligands knock out cell lines need to be used, to confirm the results.

The expression of protein M2.2 was said to reduce the expression of MICA and MICB, HMPV/∆M2.2 is suggested to be used in virus infection experiment.

Author Response

REVIEWER#2

I have carefully read this manuscript.

HMPV is a kind of virus which easily infect children during the age of 3 to 5. The mechanism of HMPV infection and pathogenesis was poorly understood. In this manuscript, Diab et al authors reported that HMPV could evade NK cell attack by downregulated the expression of the ligands of NKG2D on host cells, such as MICA, MICB, ULBP2, and ULBP3. This manuscript also provided some clues on the mechanism of HMPV evaded from NK cell, specifically, viral protein G was involved in the downregulation of ULBP2, and protein M2.2 was involved in the downregulation of MICA, MICB.

Overally, the work is well done. I suggest acceptance of this manuscript with some minor revisions.

Some questions:

Perhaps the most surprising finding is HMPV could evade NK cell attack by downregulated the expression of several ligand of NKG2D on host cells. But here the author should describe the “mock” clearly in almost all figures.

Response: We thank the reviewer for evaluating our manuscript and we agree with the reviewer. Please see the new explanation in the revised manuscript.

The HMPV/∆G was used to infect cells, it is not clear to this reviewer why the authors used this viral strain in the first place, why not other eight viral proteins delete strain or by a screen.

Response: We have stated previously that HMPV/WT-infected cells and HMPV/ΔG-infected cells upregulate the ligand expression of the activating receptor NKp46 (ref 32 in the current study). Thus, it is important to study the relevance of other ligands in response to these recombinant viruses. Please see lines 181-182 in the manuscript.

The authors described that protein G played an important role in regulating the expression of ULBP2. ULBP2 or other NKG2D ligands knock out cell lines need to be used, to confirm the results.

Response: Our study describes well the effect of HMPV infection on NKG2D ligands’ expression. However, looking for all mechanisms responsible for the downregulation of each ligand needs further investigation. In the current study, we show a strong effect of the HMPV infection by targeting the specific ligands of the activating receptor NKG2D. and addressing the involvement of two viral proteins: G and M2.2 in this effect.

The expression of protein M2.2 was said to reduce the expression of MICA and MICB, HMPV/∆M2.2 is suggested to be used in the virus infection experiment.

Response: Of course to understand better the full mechanism of the downregulation of MICA and MICB by M2.2 protein needs further investigation. Here we describe a unique effect of the HMPV infection in targeting several ligands of the NKG2D receptor.

Round 2

Reviewer 1 Report

This is a revised version of the manuscript “Human metapneumovirus escapes NK cell recognition through the downregulation of stress-induced ligands for NKG2D“. While most of the comments have been answered by the authors. There are some minor observations that remain to be considered:

  1. Figure 4. Line 288: “These findings indicate that in response to 
infection the infected cells upregulate the transcript and the total levels of ULBP2. While HMPV, via 
its viral protein G and probably additional proteins act to downregulate the surface expression of 
ULBP2 via the lysosomal pathway”. Again, this statement is confusing for the reader because the authors are making conclusions for figures 1-4, after describing data of Figure 4. It is not clear what is referred as “ …and the total levels”? Moreover, these data are correlative regarding the role of the lysosomal pathway. Thus those data suggest more than indicate. Finally, maybe this statement should be included rather in the discussion. As a suggestion:
  • These findings indicate that, in the infected cells, HMPV upregulates the transcript of ULBP2, while downregulates the surface expression of 
 protein ULBP2. In addition, the observations that infection with delta G virus increases the protein levels of ULBP2 and that ULBP2 is further upregulated in the presence of lysosome inhibitors, suggest that HMPV, via 
its viral protein G, contribute to downregulate the surface expression of 
ULBP2, likely through the lysosomal pathway.

  1. Statistical analysis. Significance symbols must be consistent across all figures. Those in Fig. 4 are not consistent with the rest of the figures. Typically, p values are *P<0.05; **P<0.01; ***P<0.001. Using different values for the significance symbols is fine but makes it harder for the reader to analyze the data.

  1. Additional round of editing will eliminate some minor typos in the manuscript.

Author Response

Reviewer#1 Comments round 2:

This is a revised version of the manuscript “Human metapneumovirus escapes NK cell recognition through the downregulation of stress-induced ligands for NKG2D“. While most of the comments have been answered by the authors. There are some minor observations that remain to be considered:

  1. Comment: Figure 4. Line 288: “These findings indicate that in response to infection the infected cells upregulate the transcript and the total levels of ULBP2. While HMPV, via 
its viral protein G and probably additional proteins act to downregulate the surface expression of 
ULBP2 via the lysosomal pathway”. Again, this statement is confusing for the reader because the authors are making conclusions for figures 1-4, after describing data of Figure 4. It is not clear what is referred as “ …and the total levels”? Moreover, these data are correlative regarding the role of the lysosomal pathway. Thus those data suggest more than indicate. Finally, maybe this statement should be included rather in the discussion. As a suggestion:

These findings indicate that, in the infected cells, HMPV upregulates the transcript of ULBP2, while downregulates the surface expression of 
 protein ULBP2. In addition, the observations that infection with delta G virus increases the protein levels of ULBP2 and that ULBP2 is further upregulated in the presence of lysosome inhibitors, suggest that HMPV, via 
its viral protein G, contribute to downregulate the surface expression of 
ULBP2, likely through the lysosomal pathway.

Response: We thank the reviewer for evaluating our manuscript. We agree with the reviewer and we changed our interpretation in the result description to “These findings suggest that HMPV/WT-infected cells target the ULBP2 protein expression, probably via its viral protein G to lysosomal degradation”. Please see lines 277-278 in the revised manuscript.

We agree with the reviewer and the statement moved to the discussion for better clarification. Please see lines 350-356 in the revised manuscript.

  1. Comment: Statistical analysis. Significance symbols must be consistent across all figures. Those in Fig. 4 are not consistent with the rest of the figures. Typically, p values are *P<0.05; **P<0.01; ***P<0.001. Using different values for the significance symbols is fine but makes it harder for the reader to analyze the data.

Response: In all figures we chose to represent the Bonferroni corrected p values as: *P<0.01; **P<0.005; ***P<0.0001. we thank the reviewer for his comment. We corrected the p values to “*p < 0.01 and **p < 0.005”. please see line 290 in the revised manuscript.

  1. Comment: Additional round of editing will eliminate some minor typos in the manuscript.

Response: additional editing performed. Please see line 349 and line 370.